# Rectovaginal Extra-Gastrointestinal Stromal Tumors (EGISTs): A Systematic Review of the Literature and a Pooled Survival Analysis

**DOI:** 10.3390/cancers17081382

**Published:** 2025-04-21

**Authors:** Eleni Papamattheou, Ioannis Katsaros, Stavros P. Papadakos, Evangelos Lianos, Elissaios Kontis

**Affiliations:** 1First Department of Obstetrics and Gynecology, National and Kapodistrian University of Athens, Alexandra General Hospital, 11528 Athens, Greece; eleni.papamattheou@gmail.com; 2First Department of Surgery, National and Kapodistrian University of Athens, Laikon General Hospital, 11527 Athens, Greece; 3First Department of Gastroenterology, National and Kapodistrian University of Athens, Laikon General Hospital, 11527 Athens, Greece; stavrospapadakos@gmail.com; 4Department of Medical Oncology, Metaxa Cancer Hospital, 18537 Piraeus, Greece; v_lianos@msn.com; 5Department of Surgical Oncology, Metaxa Cancer Hospital, 18537 Piraeus, Greece; kontiselissaios@gmail.com

**Keywords:** extra-gastrointestinal stromal tumor, EGIST, rectovaginal septum, pelvic tumor, rare tumors, CD117, DOG-1

## Abstract

Rectovaginal EGISTs are rare mesenchymal tumors arising outside the gastrointestinal tract, representing less than 5% of all GISTs. This systematic review analyzed 40 female patients, with a mean age of 55.2 years. Common symptoms included vaginal bleeding (24.3%), palpable mass (13.5%), constipation (10.8%), and abdominal pain (8.1%), though 45.9% were asymptomatic. Diagnosis was based on imaging and immunohistochemical markers, notably CD117 and DOG-1. Surgical resection was performed in 95% of cases, primarily via local excision (61.1%). Adjuvant tyrosine kinase inhibitors (TKIs) were used in 57.5% of cases, while neoadjuvant therapy was rare. Recurrence occurred in 39.4% over a median follow-up of 40 months, with a median disease-free survival of 48 months. Although rectovaginal EGISTs are rare, surgery remains the primary treatment, with adjuvant TKI therapy improving outcomes. Given the high recurrence rates, further research is needed to optimize management strategies.

## 1. Introduction

Gastrointestinal stromal tumors (GISTs) are the most common mesenchymal tumors of the gastrointestinal (GI) tract, accounting for approximately 1–3% of all GI malignancies [1]. They are believed to originate from Cajal’s interstitial cells, which regulate GI motility [2,3]. They can occur anywhere along the GI tract but more commonly are found within the stomach (60–65% of cases) and small intestine (20–25%) [1,4].

GISTs found outside the gastrointestinal tract are called extra-gastrointestinal GISTs (EGISTs), representing less than 5% of all GISTs. They were originally described in 1999 by Miettinen et al. [5] and are found in the mesentery, omentum, retroperitoneum, vagina, pelvic region, liver, abdominal wall, and the rectovaginal septum [6,7,8]. Understanding the specific genetic background leading to EGISTs’ development is an area of ongoing research [9].Their clinical presentation is multifaceted, ranging from completely asymptomatic tumors to masses causing significant abdominal pain, bleeding, and/or obstruction depending on their location, size, and growth pattern. CT and MRI scans are the main modalities used for the diagnosis of these tumors [10].

On rare occasions, these tumors are found in the rectovaginal septum, making it challenging to determine their origin and differentiate them from exophytic rectal GISTs [11]. The rectovaginal space is an anatomical area located between the posterior wall of the vagina and the anterior wall of the rectum. It plays an essential role in pelvic anatomy, providing structural support to both the rectum and vagina. It is clinically significant because it can be a site for various conditions, including infections, tumors, and abnormalities related to either gynecological or gastrointestinal systems [12]. The aim of this study is to review the available literature on these neoplasms and assess their clinical presentation, histopathological findings, optimal treatment modalities, and follow-up data.

## 2. Materials and Methods

### 2.1. Study Protocol, Search Strategy, and Inclusion Criteria

This systematic review of the literature was conducted following the Preferred Reporting Items for Systematic Reviews and Meta-Analyses 2020 statement (PRISMA) and in line with a protocol agreed a priori by all authors [13]. This systematic review was registered at Open science with the following URL: https://osf.io/kp86v (accessed on 17 April 2025). Eligible articles were identified by a search of the Medline/PubMed bibliographical database (last search: 15 January 2025). Two independent researchers (E.P., I.K.) carried out the literature search using the following search strategy: ((gastrointestinal stromal tumor) OR (GIST)) OR ((extra-gastrointestinal stromal tumor) OR (Extra-GIST)) AND (((vaginal mass) OR (vaginal tumor)) OR (rectovaginal mass)). Moreover, to further search for potentially eligible papers, we carefully examined the reference list of identified articles for eligible studies. Case reports, case series, and case studies reporting on adult female patients with a diagnosis of EGIST of the rectovaginal space were included in this review. Language restrictions were applied (only articles in English were considered eligible), without limitations placed on the time of publication. Inclusion criteria for the present literature review included case reports, case series, and case characteristic studies concerning female patients with a diagnosis of GISTs or EGISTs of the rectovaginal space. Exophytic rectal GISTs involving the vaginal septum were excluded from this review.

### 2.2. Data Extraction

Two investigators (E.P., I.K.) working independently extracted data from each eligible study based on the title, abstract, and full text, utilizing a predefined form. The following data were collected from all studies: first author’s surname, publication year, pre-operative symptoms, diagnostic modalities, treatment approach (type of surgery, neoadjuvant or adjuvant treatment), tumor size, TNM stage, and mitotic index (according to the 8th edition of the American Joint Committee on Cancer [11]), immunohistochemical characteristics, and data related to the follow-up of the patients (overall survival, disease-free survival).

### 2.3. Statistical Analysis

Statistical analysis was performed using the statistical package SPSS 26 (IBM company, Chicago, IL, USA, 2019). Numerical data were presented by their mean values ± standard deviation (SD), whereas categorical data were presented as frequencies and valid percentages. Kaplan–Meier curves were used for the estimation of the cumulative probability of recurrence, and the log rank test was applied for their comparison. Hazard ratios (HRs) were determined using the Cox proportional hazards model and are reported with a 95% confidence interval (CI). Multivariate models included all the variables that met the proportional hazards assumption or had a *p*-value < 0.1 on univariate models. A *p*-value of <0.05 (two-tailed) was considered statistically significant.

## 3. Results

### 3.1. Literature Search, Patients’ Demographics and Symptomatology

The literature search retrieved 58 articles. Thirty-one studies met predefined inclusion criteria and were included in this systematic review [4,7,8,14,15,16,17,18,19,20,21,22,23,24,25,26,27,28,29,30,31,32,33,34,35,36,37,38,39,40]. A detailed flow chart of the study is presented in Figure 1. Included articles were published from 2004 to 2024, while most of them were published after 2011.

Overall, they reported a total of 40 female patients with a mean age of 55.2 ± 15.4 years diagnosed with EGIST of the rectovaginal septum. Reported symptoms included vaginal bleeding (24.3%), palpable mass (13.5%), constipation (10.8%), and abdominal pain (8.1%); of interest, almost half of the patients did not report any symptoms (45.9%). In most patients, the diagnosis was obtained by combination of pelvic examination and an imaging modality, such as transvaginal or rectal ultrasound, MRI, or CT. For two patients whose chief complaint was constipation, colonoscopy played a key role in the final diagnosis. Half of all patients (50.0%) had a preoperative biopsy. Of note, there were no reports of use of EUS (endoscopic ultrasound).

### 3.2. Treatment Strategy

Surgical excision of the tumor was opted for in 95.0% of cases (38 patients), while 2 patients did not undergo surgery, as one case was deemed inoperable due to tumor extent and the second one received only chemotherapy [8,38]. More often, local tumor excision was preferred (61.1%), whereas 36.1% underwent extensive surgery, including total hysterectomy and colectomy. Lymph node dissection was reported in three cases (8.8%). A transabdominal approach was reported for 12 cases (36.4%), while most patients had a transvaginal approach (54.5%), and in 9.1% of cases, both approaches were implemented. Only one case (2.5%) of minimally invasive approach was recorded. No postoperative complications were reported among the included patients.

Neoadjuvant therapy was offered to 8.6% of included women, while 57.5% received adjuvant therapy with imatinib. Five patients (12.5%), who did not receive postoperatively adjuvant therapy, eventually received chemotherapy with imatinib (three patients) or sunitinib (one patient) after developing local recurrence. Sunitinib was administered as this patient developed resistance to imatinib [28].

### 3.3. Histopathological Characteristics

Histology analyses of included patients reported a mean tumor size of 6.61 ± 3.14 cm (mean ± SD). Most specimens (61.5%) were classified as T3 according to the TNM staging system. A high-grade mitotic index (>5/50 HPF) was recorded in 63.2% of patients. Stages II and IIIb were the most often encountered, each accounting for 27.5% of included cases. Table A1 presents detailed data regarding the tumor stage of included patients. CD117, DOG-1, and vimentin were positively expressed in 100% of the cases that provided relevant data. CD34 followed in frequency, being positively expressed in 97.1% of cases. Table 1 and Table A2 present detailed histopathologic characteristics of all patients.

### 3.4. Patients’ Follow-Up, Recurrence, and Survival Data

Patients were followed for an average of 40.0 ± 61.5 months. Out of the available data, 13 patients (39.4%) experienced a recurrence. Median disease-free survival was 48 months, and the cumulative disease-free survival curve is shown in Figure 2. Age (>55 vs. <55 years old), T stage (T1, T2 vs. T3, T4), mitotic index, and stage of the disease (stages I, II vs. III, IV) did not significantly affect disease-free survival, as shown in Table 2 and Figure 3. Adjuvant treatment was significantly associated with improved disease-free survival (HR: 1.66; 95% CI 1.04–2.66; *p*-value: 0.035). There was only one death recorded at 13 months postoperatively. Analysis of overall survival was not applicable (redundant), as only one event of interest was recorded during follow-up. Detailed follow-up data of each patient are shown in Table 3.

## 4. Discussion

GISTs are the most common mesenchymal tumors of the gastrointestinal tract, and they can arise in any part of GI tract or rarely in an extra-GI site. EGISTs of the rectovaginal space are quite rare neoplasms and are often misdiagnosed [11]. Diagnosing EGISTs presents significant challenges due to their rarity and atypical anatomical location, leading to frequent misdiagnosis as myosarcomas or liposarcomas. This diagnostic difficulty arises from a general lack of awareness among surgeons, gynecologists, and pathologists regarding EGISTs and their unique characteristics. However, these tumors are similar to GISTs, in terms of their histopathological and clinical features, and appropriate management remains to be clarified, as only limited data exist describing the management of rectovaginal EGIST. To our knowledge, this is the most extensive systematic review of the literature on rectovaginal EGISTs, highlighting their histologic characteristics, clinical behavior, and optimal approach.

EGISTs are usually diagnosed during the fifth to sixth decade of life, and there is no significant predilection for any particular race or ethnicity. Our study has shown that rectovaginal EGISTs are also encountered in this age group, but there are reports that they may have an earlier age of onset [31]. After conducting a thorough literature review, there was one case report of a rectovaginal EGIST in an adolescent [28]. As far as the clinical presentation is concerned, patients with this neoplasm may present with a range of non-specific symptoms, such as abnormal vaginal bleeding, abdominal pain or distension, and even with effects of urinary urgency [44,45]. About 50% of rectovaginal EGISTs are asymptomatic and are diagnosed incidentally during imaging examinations or surgery for other pathologies.

A detailed medical history, physical examination, and evaluation of risk factors, such as a family history of GIST or other inherited syndromes, are very important for reaching a diagnosis. Physical examination usually uncovers a tough, well-defined vaginal or rectal mass with or without extension into the pelvis [4]. Computed tomography (CT) scans, magnetic resonance imaging (MRI), and ultrasound imaging constitute the main imaging modalities and are important in detecting the location, size, and characteristics of the tumor. According to the study by M. Ambrosio et al., the presence of a tumor in the pelvis or abdomen which is an incidental finding during a transvaginal or transabdominal ultrasound and does not originate from the bowel or uterus is likely to point to an EGIST [46]. However, there are no specific imaging findings indicating the development of a primary GIST in the rectovaginal septum, and the diagnosis is usually made through elimination of other diagnoses [22]. Among reported cases, the use of rectal EUS has had limited applicability. However, its implementation is crucial for accurately delineating the tumor’s origin—in the rectovaginal space or rectum—especially prior to offering radical surgery. This is particularly important when a preoperative biopsy confirms GIST, preventing unnecessary rectal resection. Additionally, rectal EUS helps to assess tumor depth, invasion, and surgical feasibility. Its accuracy is significantly reduced if performed after a transvaginal excision biopsy, as anatomical planes may be distorted [47]. Therefore, rectal EUS should be incorporated early on the diagnostic pathway to optimize treatment planning and avoid misclassification, thus enhancing surgical decision making for these rare tumors.

There are three main histological patterns of gastrointestinal stromal tumors: spindle cell type (70%), epithelioid cell type (20%), and mixed type (10%) [48]. The immunochemistry of EGISTs is an important feature of these tumors and differentiates them from other mesenchymal tumors, such as leiomyoma, leiomyosarcoma, schwannoma, local extension of a primary retroperitoneal liposarcoma, benign and malignant vascular tumors, intra-abdominal fibromatosis, carcinoids with a spindle cell morphology, and metastatic disease. The typical panel of positive immunohistochemical staining includes CD117, DOG1, CD34, SMA, S100, and desmin [4]. The positive expression of CD117 and DOG1 is sufficient to confirm the diagnosis of an EGIST, and CD34 plays a crucial role in case one of the previous markers is negative [44]. All reported cases in this study demonstrated a positive expression of CD117, DOG-1, vimentin, and CD34, while most cases were characterized by spindle cell morphology on histological examination. Key immunohistochemical markers aid in distinguishing these tumors: GISTs are typically positive for CD117 and DOG-1, while leiomyomas and leiomyosarcomas express smooth muscle actin (SMA) and desmin but lack CD117. Schwannomas are marked by S100 protein positivity and absence of CD117 and DOG-1. Given their overlapping features, precise immunophenotyping is essential for accurate diagnosis and appropriate management, as GISTs require distinct therapeutic approaches [49].

In addition, GISTs are characterized by mutations in the KIT or PDGFRA genes, leading to constitutive activation of tyrosine kinase receptors, which drive uncontrolled cellular proliferation and tumor progression. The majority of KIT mutations occur in exon 11, which encodes the juxtamembrane domain and is associated with a higher rate of response to imatinib therapy. [11]. However, mutations in exon 9, found in the extracellular domain, are more common in EGISTs compared to GISTs, and these tumors often demonstrate a primary resistance to imatinib, necessitating alternative therapeutic strategies such as sunitinib or avapritinib for effective disease control [44]. Additionally, PDGFRA mutations are frequently observed in gastric GISTs but occur at lower rates in EGISTs. These mutations typically involve exon 18, leading to the substitution of aspartic acid with valine at codon 842 (D842V), which is associated with primary resistance to imatinib. However, newer-generation tyrosine kinase inhibitors, such as avapritinib, have shown promising efficacy in these cases [11]. Other mutations in genes such as BRAF, primarily involving exon 15 (V600E), have been identified in a smaller subset of GISTs and are linked to imatinib resistance [50]. Similarly, NF1 gene mutations are commonly seen in neurofibromatosis type 1-associated GISTs and are characterized by a distinct molecular profile lacking KIT or PDGFRA mutations [1]. Furthermore, SDH-deficient GISTs (linked to mutations in SDHA, SDHB, SDHC, or SDHD) are a unique subset of tumors that are often resistant to conventional TKI therapy [51].The identification of these genetic alterations has profound clinical implications, as it enables molecularly driven treatment selection and personalized therapy. As targeted agents such as avapritinib, ripretinib, and regorafenib continue to evolve, molecular profiling of EGISTs and GISTs is crucial for optimizing patient outcomes in the era of precision oncology [2].

The current standard therapy of these neoplasms is en-block surgical resection of the tumor with negative surgical margins, while avoiding intraoperative tumor rupture is of major importance. The possibility of lymph node metastasis is rare, so lymphadenectomy is not recommended. It is quite challenging to achieve R0 resections with no residual tumor due the anatomy of the rectovaginal space; thus, high local recurrence rates have been reported [44]. In this study, the open surgical approach, either transabdominal or transvaginal, was the preferred method of tumor removal. According to the results of Hiroshi Ohtani’s meta-analysis, a minimally invasive approach can be implemented for this tumor without compromising oncological outcomes, while allowing for reduced operative time, length of hospitalization, and postoperative morbidity [52]. On the other hand, the laparoscopic approach is discouraged in patients who have large tumors because of the high risk of tumor rupture, which is associated with a very high risk of recurrence (15–20%) [11].

The characteristics of a tumor, including stage, size, mitotic count, location and intraoperative tumor rupture, are well-documented prognostic factors for adjuvant therapy with TKIs, such as imatinib mesylate. They are mainly given postoperatively and are continued for up to 3 years to decrease the risk of recurrence [50]. According to the National Comprehensive Cancer Network and European Society of Medical Oncology, neoadjuvant treatment with imatinib and avapritinib is an option for cases of metastatic disease, locally advanced disease which is inoperable, or if upfront surgery is unlikely to achieve R0 resection [53,54]. In cases of response to TKIs, surgical resection should follow, but treatment with imatinib mesylate will be continued postoperatively [11,54]. The rarity of rectovaginal EGISTs may contribute to a lack of standardized protocols for neoadjuvant TKI use. Moreover, these tumors are often diagnosed at a stage where surgical resection is feasible without downsizing. In cases where neoadjuvant therapy was offered, mutation testing to guide treatment selection was not necessarily performed, thus potentially limiting its expected effectiveness.

Rectovaginal EGISTs tend to have more aggressive clinical behavior compared to GISTs, probably due to lower rates of R0 resection given their challenging location [7]. Usually, these tumors show local recurrence and metastasize to the liver, the mesentery and omentum, lung, subcutaneous tissues, lymph nodes, or bone [11]. Known prognostic factors include the mitotic rate, tumor size, tumor site, and tumor rupture. More specifically, the mitotic rate affects the speed at which recurrence appears. High-risk patients have a relapse within 1–3 years from the end of adjuvant therapy [54]. EGISTs found at other sites have also shown similar biological behavior. Padhi et al. conducted a review of 19 pancreatic EGISTs and reported a recurrence rate of 26.3% occurring between 5 and 66 months after surgery, with only R0 resection showing a statistically significant association with recurrence [55]. Reith et al. analyzed 48 abdominal and retroperitoneal EGISTs and showed that high mitotic activity and tumor necrosis were risk factors for local recurrence [56]. Our study has shown that approximately 4 out of 10 patients had recurrence occurring an average of 4 years after index surgery. Five percent of patients within this cohort developed distant metastasis within the liver [20]. All patients who presented with recurrence received adjuvant therapy with imatinib.

As expected, our study has several limitations. The data primarily originated from retrospective cases, relying on the accuracy of record keeping at individual centers. Additionally, the data exhibited significant heterogeneity, as they were collected from multiple centers employing diverse treatment. The rarity of rectovaginal EGISTs resulted in a small sample size, which may limit the study’s ability to detect potential associations between recorded variables, possibly leading to false-negative findings. Most studies did not report on margin status, making such an analysis unfeasible. Genetic profiling was also inconsistently documented, and immunohistochemical markers were mostly positive in reported cases, with no available control group for direct comparison. Despite these challenges, this study contributes valuable insights and helps to address the existing knowledge gap in understanding this rare condition.

## 5. Conclusions

In conclusion, rectovaginal EGISTs are exceptionally rare tumors with diverse clinical presentations, often resulting in delayed diagnosis. Despite their unusual location, they share histopathological and immunohistochemical characteristics with gastrointestinal GISTs, with CD117 and DOG-1 as key diagnostic markers. Imaging techniques and histopathology remain essential for accurate diagnosis. Surgical resection remains the cornerstone of treatment, with the primary objective of achieving negative margins while minimizing the risk of tumor rupture. However, the complex anatomy of the rectovaginal space makes achieving clear surgical margins difficult, leading to a higher recurrence rate. Adjuvant tyrosine kinase inhibitors, such as imatinib, are crucial for recurrence prevention, particularly in high-risk patients. Neoadjuvant therapy may increase resectability among selected patients, though its utilization is limited. Recurrence affects nearly 40% of patients within four years, highlighting the necessity of long-term surveillance. Future research should aim to define optimal treatment protocols, including the role of neoadjuvant and adjuvant therapies, with prospective studies and molecular analyses guiding personalized approaches to improve outcomes for this rare and challenging tumor type.

## Figures and Tables

**Figure 1 cancers-17-01382-f001:**
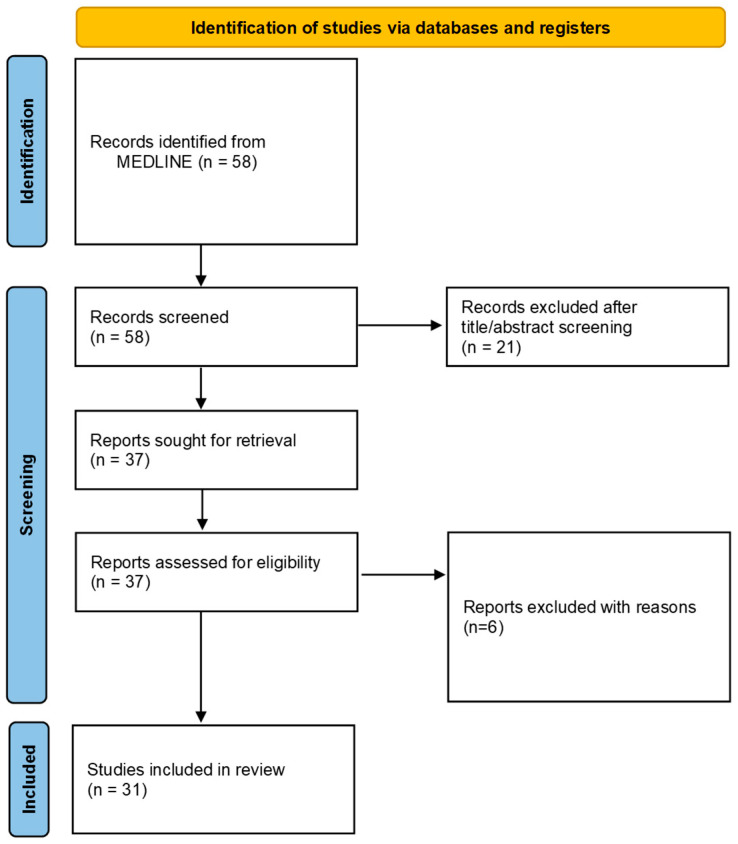
Flow chart of this systematic review.

**Figure 2 cancers-17-01382-f002:**
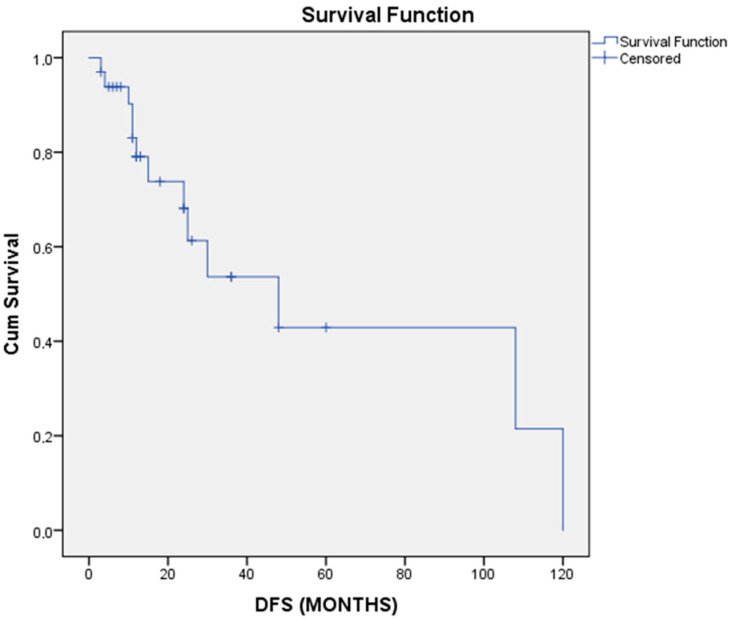
Cumulative disease-free survival (DFS) Kaplan–Meier curve of recto-vaginal EGISTs.

**Figure 3 cancers-17-01382-f003:**
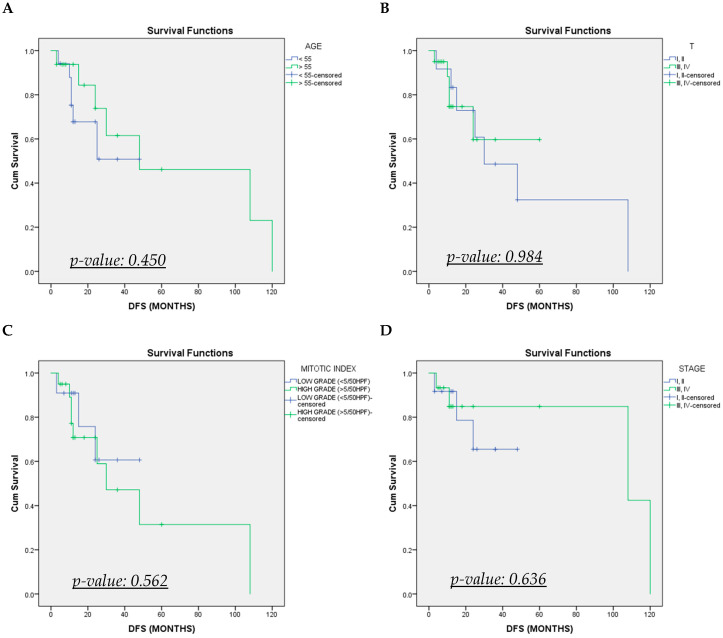
Cumulative disease-free survival (DFS) Kaplan–Meier curves of recto-vaginal EGISTs stratified by (**A**) age, (**B**) T-stage, (**C**) mitotic index, and (**D**) stage.

**Table 1 cancers-17-01382-t001:** Histopathological characteristics of included patients.

Study ID	No. of Mitosis per 50 HPF	Histological Grade
Nasu et al. (2004) [30]	1–2	LOW
Ceballos et al. (2004) [17]	12–15	HIGH
E H Weppler et al. (2005) [38]	5	LOW
Senzan HSU et al. (2006) [21]	>10	HIGH
Sennzan HSUet al. (2006) [21]	>5	HIGH
Takano et al. (2006) [36]	1–2	LOW
Maggie M Lam et al. (2006) [23]	15	HIGH
Maggie M Lam et al. (2006) [23]	12	HIGH
Maggie M Lam et al. (2006) [23]	16	HIGH
Youn-Jeong Kim et al. (2007) [22]	5	LOW
S Nagase et al. (2007) [29]	1	LOW
S Nagase et al. (2007) [29]	2–3	LOW
Isela Molina et al. (2009) [41]	25	HIGH
Josefa Marcos Sanmartín et al. (2009) [32]	16	HIGH
Wenjing Zhang et al. (2009) [39]	10	HIGH
Teresa Blasco Segura et al. (2010) [33]	16	HIGH
A Frilling et al. (2010) [20]	NR	NR
José Humberto Tavares Guerreiro Fregnani et al. (2011) [19]	4	LOW
Antje-Friederike Pelz et al. (2011) [31]	10	HIGH
Julio Vázquez et al. (2012) [37]	10	HIGH
Mario Munoz et al. (2013) [28]	40	HIGH
Mario Munoz et al. (2013) [28]	50	HIGH
Abbas Agaimy et al. (2013) [14]	14	HIGH
Abbas Agaimy et al. (2013) [14]	10	HIGH
Abbas Agaimy et al. (2013) [14]	11	HIGH
Marcos N Meléndez et al. (2014) [27]	1–3	LOW
Y H Lee et al. (2015) [25]	4–5	LOW
Qiu-Yu Liu et al. (2016) [26]	25	HIGH
Wissam Hanayneh (2018) [8]	45	HIGH
Wissam Hanayneh (2018) [8]	4	LOW
Brian H Le et al. (2019) [24]	3–4	LOW
Min Cheng et al. (2019) [4]	7	LOW
David Carney et al. (2019) [16]	30	LOW
Jen Sothornwit et al. (2020) [35]	0–1	LOW
J Shi et al. (2020) [34]	>5	HIGH
Shuai Liu et al. (2021) [42]	14	HIGH
Reo Ando et al. (2022) [15]	2–4	LOW
Ying Liu et al. (2022) [43]	3	LOW
Ying Liu et al. (2022) [43]	3	LOW
Papamattheou et al. (2024) [40]	>20	HIGH

HPF: high-power field.

**Table 2 cancers-17-01382-t002:** Univariate Cox proportional analysis regarding disease-free survival of rectovaginal EGISTs.

	HR	95% CI	*p*-Value
Age > 55 years	0.63	0.18–2.14	0.452
T1–2 vs. T3–4	0.99	0.29–3.36	0.984
STAGE I–II vs. III–IV	0.72	0.30–1.73	0.460
High Mitotic Index (>5/50 HPF)	1.48	0.39–5.61	0.563
Neoadjuvant TKIs	1.96	0.24–16.27	0.528
Adjuvant TKIs	**1.66**	**1.04–2.66**	**0.035**

HR: hazard ratio; 95% CI: 95% confidence interval; HPF: high-power field; TKI: tyrosine kinase inhibitors.

**Table 3 cancers-17-01382-t003:** Treatment approach of included cases and follow-up data.

Study	Surgery	Neoadjuvant	Adjuvant	Follow-Up (Months)
Nasu et al. (2004) [30]	TAH and LN	NO	NO	13 NED
Ceballos et al. (2004) [17]	LE	NO	NO	recurrence at 108 m
E H Weppler et al. (2005) [38]	-	YES (imatinib)	NO	NR
Senzan HSU et al. (2006) [21]	TAH + BSO + LN + RR	NO	YES (imatinib)	24 NED
Sennzan HSUet al. (2006) [21]	TAH + BSO + RR	NO	YES (imatinib)	6 NED
Takano et al. (2006) [36]	ES	NO	NO	12 NED
Maggie M Lam et al. (2006) [23]	NR	NR	NR	recurrence at 12 m
Maggie M Lam et al. (2006) [23]	NR	NR	NR	recurrence at 10 m
Maggie M Lam et al. (2006) [23]	NR	NR	NR	NR
Youn-Jeong Kim et al. (2007) [22]	TAH + BSO	NO	YES (imatinib)	7 NED
S Nagase et al. (2007) [29]	LE	NO	NO	48 NED
S Nagase et al. (2007) [29]	LE	NO	YES, after recurrence	recurrence at 3 m and 3 NED
Isela Molina et al. (2009) [41]	LE	NO	YES (imatinib and radiation)	recurrence at 48 m and 60 m and18 NED
Josefa Marcos Sanmartín et al. (2009) [32]	LE	NO	YES (imatinib)	NED 12
Wenjing Zhang et al. (2009) [39]	ES	NO	NO	NED 11
Teresa Blasco Segura et al. (2010) [33]	LE	NO	YES	NR
A Frilling et al. (2010) [20]	LE	NO	YES (imatinib and chemotherapy)	recurrence at 108 m and 120 AWD
José Humberto Tavares Guerreiro Fregnani et al. (2011) [19]	LE	NO	YES (imatinib, after recurrence)	recurrence at 15 m and 6 m NED
Antje-Friederike Pelz et al. (2011) [31]	ES	NO	YES (imatinib after recurrence)	recurrence 11, 23, 36 and 8 m NED
Julio Vázquez et al. (2012) [37]	LE	NO	YES (imatinib)	NED 12
Mario Munoz et al. (2013) [28]	LE	NO	YES (imatinib and sunitinib after recurrence)	recurrence at 4, 8, 13 and DOD at 13 m
Mario Munoz et al. (2013) [28]	LE	NO	YES (imatinib, after recurrence)	recurrence at 82 m
Abbas Agaimy et al. (2013) [14]	LE	YES	YES	recurrence at 30 m and 10 m ANED
Abbas Agaimy et al. (2013) [14]	LE	NO	YES (imatinib)	recurrence at 11, 23, 36 and 44 m ANED
Abbas Agaimy et al. (2013) [14]	ES	NO	NO	NR
Marcos N Meléndez et al. (2014) [27]	LE	NO	YES (imatinib after recurrence)	recurrence at 24 m and 22 m NED
Y H Lee et al. (2015) [25]	ES	NO	YES (imatinib)	26 m NED
Qiu-Yu Liu et al. (2016) [26]	LE	NO	YES (imatinib)	5 m NED
Wissam Hanayneh (2018) [8]	TAH and BSO	NO	YES (imatinib)	8 m NED
Wissam Hanayneh (2018) [8]		YES	NR	3 m NED
Brian H Le et al. (2019) [24]	LE	NO	NO	24 m NES
Min Cheng et al. (2019) [4]	TAH and BSO	NO	YES (imatinib)	36 m NED
David Carney et al. (2019) [16]	LE	YES	NR	NR
Jen Sothornwit et al. (2020) [35]	TAH	NO	NO	11 m NED
J Shi et al. (2020) [34]	LE	NO	YES (imatinib)	13 m NED
Shuai Liu et al. (2021) [42]	LE	NO	NO	60 m NED
Reo Ando et al. (2022) [15]	LE	NO	YES (imatinib after delivery)	36 m NED
Ying Liu et al. (2022) [43]	LE	NR	NR	NR
Ying Liu et al. (2022) [43]	ES	NR	NR	recurrence at 161 m
Papamattheou et al. (2024) [40]	LE and liver metastasectomy	NO	YES (imatinib)	18 m NED

TAH: total abdominal hysterectomy, BSO: bilateral salpingo-oophorectomy, LN: lymphadenectomy, LE: local excision, NR: not reported, RR: rectal resection, ES: extensive surgery, NED: no evidence of disease, AWD: alive with disease, DOD: died of disease.

## Data Availability

All available data are presented in the tables within the article.

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
