# Peer review of "Rectovaginal Extra-Gastrointestinal Stromal Tumors (EGISTs): A Systematic Review of the Literature and a Pooled Survival Analysis"

_cancers, 2025, doi:10.3390/cancers17081382_

Round 1

Reviewer 1 Report

Comments and Suggestions for Authors

Thank you for this work related to a rare tumour and one that may not be considered in this space.

The work is well organized and written.  As stated in the paper, the work is limited by the the low numbers and retrospective nature of the study, making it difficult to determine how to use the information obtained from this review.  As there is no survival difference with any of the tumor stages or grades, it would be interesting to know how recto-vaginal eGISTs compare to eGISTs in other areas.

The discussion comments on the role of margins and molecular analysis in the treatment of GISTs.  The paper would be strengthened if the margin status and molecular profiling could be built into the analysis.

Author Response

Comment 1: Thank you for this work related to a rare tumour and one that may not be considered in this space.
The work is well organized and written.  As stated in the paper, the work is limited by the the low numbers and retrospective nature of the study, making it difficult to determine how to use the information obtained from this review.  As there is no survival difference with any of the tumor stages or grades, it would be interesting to know how recto-vaginal eGISTs compare to eGISTs in other areas.
Response1: Thank you for your comments. We added a comparison with pancreatic and abdominal E-GISTs in the Discussion section highlighting similarities in clinical behavior.

Comment 2: The discussion comments on the role of margins and molecular analysis in the treatment of GISTs.  The paper would be strengthened if the margin status and molecular profiling could be built into the analysis.

Response 2: Thank you for your input. Most studies did not report margin status, making such an analysis unfeasible. Additionally, genetic profiling was inconsistently documented, and immunohistochemical markers (DOG-1, CD117, vimentin) were uniformly positive in reported cases, with no available control group for comparison. Moreover, PDGFRA data were available for only three cases. These limitations have been acknowledged in the respective section.

Reviewer 2 Report

Comments and Suggestions for Authors

Reviewer Report

Manuscript ID: cancers-3529367

Title: Rectovaginal extra-gastrointestinal stromal tumors (E-GISTs): a systematic review of the literature and a pooled survival analysis

This is a timely and well-structured systematic review on a rare entity — rectovaginal E-GISTs. The article has both clinical relevance and academic merit and fills a significant knowledge gap. The authors have adhered well to PRISMA guidelines and presented robust pooled data. However, a few points need clarification and refinement for optimal clarity and impact.

Major Comments:

(a) Definition and Classification of EGISTs

  • Please clarify early in the manuscript whether all included cases represent true EGISTs (without gastrointestinal wall attachment) or if some were GISTs with extraluminal extension. This is a known controversy in the literature.

(b) Use of Rectal EUS

  • The manuscript mentions the absence of EUS in reported cases and highlights its potential diagnostic utility. This point deserves stronger emphasis in the discussion as a practical recommendation for future diagnostic workups.

(c) Neoadjuvant Therapy

  • Since neoadjuvant TKIs are rarely used, the rationale for their underuse should be discussed. Were tumors diagnosed too late or considered inoperable upfront? Were mutational analyses performed pre-treatment?

(d) Heterogeneity and Limitations

  • A more detailed paragraph on limitations is warranted. For instance, retrospective nature, heterogeneity in surgical approaches, incomplete data on mutation status, and lack of centralized pathology review.

(e) Prognostic Factors – Interpretation of Non-Significant Findings

  • The manuscript shows no statistically significant difference in DFS based on age, stage, or mitotic index. The authors should briefly hypothesize why this might be (e.g., small sample size, variable adjuvant therapy).

Minor Comments:

  • Abstract: Please include the number of included studies (n=31) in the abstract for completeness.
  • Language Editing: Minor typographical and grammatical errors are present. A light proofreading round is suggested for polishing the text.
  • Reference Formatting: Some citations in the text (e.g., [11]) appear multiple times for different contexts — please double-check alignment between citation and reference content.

Suggestions for Further Enrichment:

  • Add a “Clinical Implications” box or subsection summarizing diagnostic and treatment takeaways for gynecologists, oncologists, and surgeons.
  • Consider adding a table comparing EGISTs of the rectovaginal septum vs EGISTs from other pelvic sites or GISTs in general, if such data are available from literature.
Comments on the Quality of English Language

Minor typographical and grammatical errors are present. A light proofreading round is suggested for polishing the text.

Author Response

Comment 1: This is a timely and well-structured systematic review on a rare entity — rectovaginal E-GISTs. The article has both clinical relevance and academic merit and fills a significant knowledge gap. The authors have adhered well to PRISMA guidelines and presented robust pooled data. However, a few points need clarification and refinement for optimal clarity and impact.

Major Comments:
(a) Definition and Classification of EGISTs
•    Please clarify early in the manuscript whether all included cases represent true EGISTs (without gastrointestinal wall attachment) or if some were GISTs with extraluminal extension. This is a known controversy in the literature.
Response 1: Thank you for your comments. We have now clarified this issue in the Introduction. Only cases with no attachment to the gastrointestinal wall were classified as EGISTs, distinguishing them from exophytic GISTs.

Comment 2: 
(b) Use of Rectal EUS
•    The manuscript mentions the absence of EUS in reported cases and highlights its potential diagnostic utility. This point deserves stronger emphasis in the discussion as a practical recommendation for future diagnostic workups.

Response 2: Thank you for your input. We have expanded the discussion on the role of rectal Given its ability to delineate tumor origin with high accuracy, we recommend incorporating EUS into routine preoperative assessment, particularly in cases where the anatomical distinction between rectal and rectovaginal tumors is unclear. This could aid in treatment planning and reduce unnecessary radical surgery.

Comment 3: 
(c) Neoadjuvant Therapy
•    Since neoadjuvant TKIs are rarely used, the rationale for their underuse should be discussed. Were tumors diagnosed too late or considered inoperable upfront? Were mutational analyses performed pre-treatment?

Response 3: Thank you for your comment. Neoadjuvant TKIs are an option in cases of metastatic disease, advanced disease which is deemed inoperable, or if upfront surgery is unlikely to achieve R0 status. The rarity of rectovaginal EGISTs may contribute to a lack of standardized protocols for neoadjuvant TKI use. We acknowledge these in the relevant Discussion section.

Comment 4:
(d) Heterogeneity and Limitations
•    A more detailed paragraph on limitations is warranted. For instance, retrospective nature, heterogeneity in surgical approaches, incomplete data on mutation status, and lack of centralized pathology review.
Response 4: Thank you for your comment. We further expanded the limitations section.

Comment 5: 
(e) Prognostic Factors – Interpretation of Non-Significant Findings
•    The manuscript shows no statistically significant difference in DFS based on age, stage, or mitotic index. The authors should briefly hypothesize why this might be (e.g., small sample size, variable adjuvant therapy).

Response 5: Thank you for your input. We acknowledged these findings in discussion and limitations section. The small sample size may limit the study's ability to detect potential associations.

Comment 6: 
•    Abstract: Please include the number of included studies (n=31) in the abstract for completeness.
Response 6: We included the total number studies.

Comment 7: 
•    Language Editing: Minor typographical and grammatical errors are present. A light proofreading round is suggested for polishing the text.
Response 7: Thank you for your input. A proofread by a native English speaker and subsequent errors corrections was made.

Comment 8:
•    Reference Formatting: Some citations in the text (e.g., [11]) appear multiple times for different contexts — please double-check alignment between citation and reference content.
Response 8: Thank you for your comments. We used Endnote reference manager to correct the references.

Reviewer 3 Report

Comments and Suggestions for Authors

Dear authors,

Thanks very much for your work

The paper is well-designed and written. It was a bit difficult to find comments.

1- The main concern is why these are considered extra GIT? Why were they not considered exophytic from the anterior rectal wall as GIST usually presents with exophytic masses?

2- please mention the response pattern to neoadjuvant therapy in the patient who received it

3- What was the rationale for receiving chemotherapy in the patient who received it?

4- The conclusion is too long. Please reduce it to half at most

Author Response

Comment 1: 
Dear authors,
Thanks very much for your work
The paper is well-designed and written. It was a bit difficult to find comments.
1- The main concern is why these are considered extra GIT? Why were they not considered exophytic from the anterior rectal wall as GIST usually presents with exophytic masses?
Response 1: Thank you for your comments. Histology revealed that these tumors did not derive from the rectum and thus were true E-GISTs. We further clarified in methods section that exophytic rectal GISTs were excluded from this review.
Comment 2:
2- please mention the response pattern to neoadjuvant therapy in the patient who received it
Response 2:
Thank you for your input. Neoadjuvant treatment was rarely used and included cases did not specified response according to RECIST criteria. We acknowledge this fact in the limitation section.

Comment 3:
3- What was the rationale for receiving chemotherapy in the patient who received it?
Response 3:
Thank you for your input. This is a systematic review of available cases, and we did not intervene with the management of the included patients. No definite guidelines exist for the management of E-GISTs and management mainly is extrapolated from data of GIST management. We further analyzed this fact in the Discussion section.
Comment 4:
4- The conclusion is too long. Please reduce it to half at most
Response 4:
Thank you for your comment. We revised the conclusion section to be more concrete. 

Reviewer 4 Report

Comments and Suggestions for Authors

This systematic review provides valuable insights into rectovaginal extra-gastrointestinal stromal tumors (EGISTs), a rare entity with limited available literature. The study is well-structured, with a comprehensive literature search and clear methodology. However, several aspects require improvement:

  1. The study’s novelty should be explicitly highlighted, particularly how it differs from previous systematic reviews. The clinical relevance of the findings should be further emphasized, including implications for surgical management and tyrosine kinase inhibitor (TKI) therapy.
  2. There is no mention of PROSPERO registration or another registry, which is critical for systematic reviews to ensure transparency. If registered, please include the registration number. If not, provide a justification.
  3. The study lacks a risk of bias (RoB) assessment, which is essential in systematic reviews. Consider using ROBINS-I or Cochrane RoB tools to evaluate potential biases in the included studies.
  4. While Kaplan-Meier curves are presented, Cox proportional hazards regression should be included to assess independent predictors of recurrence and survival.
  5. The discussion on KIT and PDGFRA mutations in EGISTs should be expanded. A comparison with classical GISTs regarding molecular features and targeted therapy response would strengthen the manuscript.
  6. Tables 1 and 2 contain extensive information, reducing readability. Consider moving detailed data to supplementary materials and presenting only key findings in the main text.
Comments on the Quality of English Language

The overall English quality is acceptable, but several sentences could be more concise and precise.

Some sections (e.g., Discussion) contain redundant phrasing, which should be streamlined for clarity.

Minor grammatical errors and awkward sentence structures need revision. A professional language edit is recommended.

Author Response

Comment 1:
This systematic review provides valuable insights into rectovaginal extra-gastrointestinal stromal tumors (EGISTs), a rare entity with limited available literature. The study is well-structured, with a comprehensive literature search and clear methodology. However, several aspects require improvement:
The study’s novelty should be explicitly highlighted, particularly how it differs from previous systematic reviews. The clinical relevance of the findings should be further emphasized, including implications for surgical management and tyrosine kinase inhibitor (TKI) therapy.
Response 1: Thank you for your input. This is the most conclusive review of the literature regarding rectovaginal E-GISTs and we analyzed their clinical behavior and optimal management. 

Comment 2: 
There is no mention of PROSPERO registration or another registry, which is critical for systematic reviews to ensure transparency. If registered, please include the registration number. If not, provide a justification.
Response 2: Thank you for your comment. We registered this protocol at Open Science database and added this to Methods section.

Comment 3:
The study lacks a risk of bias (RoB) assessment, which is essential in systematic reviews. Consider using ROBINS-I or Cochrane RoB tools to evaluate potential biases in the included studies.
Response 3: Thank you for your input. These risk of bias tools are not usually used for case reports, as they present a single case.

Comment 4:
While Kaplan-Meier curves are presented, Cox proportional hazards regression should be included to assess independent predictors of recurrence and survival.
Response 4: 
Thank you for your comment. We conducted cox analysis of the collected data and present them at Results section and a new Table.

Comment 5:
The discussion on KIT and PDGFRA mutations in EGISTs should be expanded. A comparison with classical GISTs regarding molecular features and targeted therapy response would strengthen the manuscript.
Response 5: Thank you for your input. We expanded the relevant discussion section.

Comment 6:
Tables 1 and 2 contain extensive information, reducing readability. Consider moving detailed data to supplementary materials and presenting only key findings in the main text.
Response 6: Thank you for your input. We revised the tables to improve readability and maintain focus on critical aspects of the study.

Round 2

Reviewer 4 Report

Comments and Suggestions for Authors

This is a well-conducted and substantially improved systematic review on rectovaginal EGISTs. The addition of Cox regression analysis, detailed molecular discussion, and study registration has clearly strengthened the manuscript. The authors are to be commended for their comprehensive work on this rare and clinically challenging entity.

To further improve the clarity and presentation of the manuscript, we offer the following minor suggestions:

  1. While the English has improved, some sections remain overly verbose, with long or repetitive sentences. We recommend reviewing the manuscript for concise expression and removing redundancies—especially in the Introduction and Discussion sections. A final professional language edit would further enhance readability.

  2. Tables 1 and 3 contain extensive raw data, which, while valuable, can reduce readability in the main text. Consider moving the full tables to the Supplementary Materials and providing simplified summary tables in the main manuscript to improve accessibility for readers.

Comments on the Quality of English Language

While the English has improved, some sections remain overly verbose, with long or repetitive sentences. We recommend reviewing the manuscript for concise expression and removing redundancies—especially in the Introduction and Discussion sections. A final professional language edit would further enhance readability.

Author Response

Comment 1: While the English has improved, some sections remain overly verbose, with long or repetitive sentences. We recommend reviewing the manuscript for concise expression and removing redundancies—especially in the Introduction and Discussion sections. A final professional language edit would further enhance readability.

Response 1: A professional English language editor has reviewed the text, and necessary improvements have been made, particularly in the Introduction and Discussion sections, to enhance readability.

Comment 2: Tables 1 and 3 contain extensive raw data, which, while valuable, can reduce readability in the main text. Consider moving the full tables to the Supplementary Materials and providing simplified summary tables in the main manuscript to improve accessibility for readers.

Response 2: Supplementary Tables A1 and A2 are added and in the main manuscript we included simplified tables to improve accessibility for readers.